# Functional Repercussions of Hypoxia-Inducible Factor-2α in Idiopathic Pulmonary Fibrosis

**DOI:** 10.3390/cells11192938

**Published:** 2022-09-20

**Authors:** Ana Karen Torres-Soria, Yair Romero, Yalbi I. Balderas-Martínez, Rafael Velázquez-Cruz, Luz Maria Torres-Espíndola, Angel Camarena, Edgar Flores-Soto, Héctor Solís-Chagoyán, Víctor Ruiz, Ángeles Carlos-Reyes, Citlaltepetl Salinas-Lara, Erika Rubí Luis-García, Jaime Chávez, Manuel Castillejos-López, Arnoldo Aquino-Gálvez

**Affiliations:** 1Red MEDICI, Carrera de Médico Cirujano, Facultad de Estudios Superiores de Iztacala Universidad Nacional Autónoma de México, Mexico City 54090, Mexico; 2Facultad de Ciencias, Universidad Nacional Autónoma México, Mexico City 04510, Mexico; 3Laboratorio de Biología Computacional, Instituto Nacional de Enfermedades Respiratorias Ismael Cosío Villegas, Mexico City 14080, Mexico; 4Laboratorio de Genómica del Metabolismo Óseo, Instituto Nacional de Medicina Genómica, Mexico City 14610, Mexico; 5Laboratorio de Farmacología, Instituto Nacional de Pediatría, Mexico City 04530, Mexico; 6Laboratorio de HLA, Instituto Nacional de Enfermedades Respiratorias Ismael Cosío Villegas, Mexico City 04530, Mexico; 7Departamento de Farmacología, Facultad de Medicina, Universidad Nacional Autónoma de México, Mexico City 04510, Mexico; 8Subdirección de Investigaciones Clínicas, Instituto Nacional de Psiquiatría Ramón de la Fuente Muñiz, Mexico City 14370, Mexico; 9Departamento de Fibrosis Pulmonar, Laboratorio de Biología Molecular, Instituto Nacional de Enfermedades Respiratorias Ismael Cosío Villegas, Mexico City 14080, Mexico; 10Laboratorio de Onco-Inmunobiología, Departamento de Enfermedades Crónico-Degenerativas, Instituto Nacional de Enfermedades Respiratorias Ismael Cosío Villegas, Mexico City 14080, Mexico; 11Departamento de Fibrosis Pulmonar, Laboratorio de Biología Celular, Instituto Nacional de Enfermedades Respiratorias Ismael Cosío Villegas, Mexico City 14080, Mexico; 12Departamento de Hiperreactividad Bronquial, Instituto Nacional de Enfermedades, Respiratorias Ismael Cosío Villegas, Mexico City 14080, Mexico; 13Departamento de Epidemiología y Estadística, Instituto Nacional de Enfermedades, Respiratorias Ismael Cosío Villegas, Mexico City 14080, Mexico

**Keywords:** lung regeneration, IPF, HIFs, hypoxia

## Abstract

Hypoxia and hypoxia-inducible factors (HIFs) are essential in regulating several cellular processes, such as survival, differentiation, and the cell cycle; this adaptation is orchestrated in a complex way. In this review, we focused on the impact of hypoxia in the physiopathology of idiopathic pulmonary fibrosis (IPF) related to lung development, regeneration, and repair. There is robust evidence that the responses of HIF-1α and -2α differ; HIF-1α participates mainly in the acute phase of the response to hypoxia, and HIF-2α in the chronic phase. The analysis of their structure and of different studies showed a high specificity according to the tissue and the process involved. We propose that hypoxia-inducible transcription factor 2a (HIF-2α) is part of the persistent aberrant regeneration associated with developing IPF.

## 1. Introduction

Idiopathic pulmonary fibrosis (IPF) has a poor prognosis, with a median survival of 24–30 months [1,2], and is characterized by reduced functional capacity, dyspnea, and hypoxia induced by exercise or at rest [3,4,5]. Destruction of lung architecture impairs gas exchange and progresses to hypoxic respiratory failure, a hallmark of advanced disease [2]. The blood oxygen saturation level is considered an important parameter because its decrease during endurance tests predicts survival in patients with IPF [6]. Most patients have a poor quality of life due to low physical activity and limited exercise tolerance [2,7], decreased lung compliance leading to mechanical ventilation, and increased respiratory muscle energy expenditure, driving dyspnea [8,9].

Exertional dyspnea and worsening hypoxia are clinical features of IPF, and no drug is available to treat these two symptoms [10]; even oxygen inhalation does not improve tolerance to physical exertion in most patients and, therefore, does not relieve shortness of breath [11,12]. Obstructive sleep apnea (OSA) is known to be a risk factor for IPF; intermittent hypoxia (HI) and reoxygenation of OSA contribute to a poor prognosis [13]. Chronic exposure to HI increases mortality, lung inflammation, and pulmonary fibrosis in BLM-treated mice, suggesting a worse prognosis in patients who have IPF and severe OSA [13,14]. Most in vitro studies with cells and in vivo with animal models have shown that hypoxia is a determining factor in the progression and development of the disease. However, at the clinical level, there are still several questions.

The mechanisms by which hypoxia and transcription factors are involved have not been fully described. Initially, the relationship is directly proportional, since a higher degree of hypoxia is observed when the extracellular matrix accumulates. Notably, this effect was observed in fibroblasts, which are the cells in charge of the matrix remodeling; however, in light of recent discoveries in the pathophysiology of IPF, we suggest that the role of hypoxia has an impact beyond this.

IPF is an age-related and chronic lung disease characterized by alteration of the typical structure of the lung and progressive loss of respiratory capacity, whose etiology has not yet been elucidated [15]. IPF patients have few therapeutic options for antifibrotic drugs, which have shown limited efficacy, such as Pirfenidone and Nintedanib [16]. 

The physiopathology of IPF has been proposed as epithelial-drive fibrosis, with converging genetic and environmental factors [17]. At an early stage, lung injury is observed, with an aberrant response of epithelial cells that secrete many mediators for fibroblast migration and activation [18]. Recent evidence has shown that these epithelial cell populations, particularly a group of basaloid cells identified by single-cell RNA sequencing (scRNA-seq) and the expression of marker senescence, development, and differentiation, are critical in the early stage of fibrotic lesions [19]. These basaloid cells are lined with myofibroblasts activated in a complex microenvironment where hypoxia, through HIFs (HIF-1α, -2α, and -3α), could be involved in the establishment of profibrotic feedback promoting the development of IPF [20,21,22,23]. In addition, it has been shown that these mesenchymal cells can be helpful in the formation of the niche for alveolar regeneration [24,25]. Endothelial cells and their mediators are also involved in this interaction within alveoli [26]. Notably, Kobayashi et al. suggest that alveolar type-2 epithelial cells require a transitional state for terminal differentiation into type 1; this pre-alveolar type-1 transitional cell state (PATS) is a persistent phenomenon associated with failed regeneration in IPF [27]. Additionally, evidence shows that transdifferentiation of epithelial cells can be induced by interaction with aberrant mesenchymal cells [28]. Therefore, the role that hypoxia may have in differentiation and the possible regulation of epithelial cell transdifferentiation is essential to elucidate, since these cells are in constant interaction with cells that are in a hypoxic microenvironment.

## 2. Lung Oxygenation and Hypoxic Conditions

The lungs are responsible for capturing oxygen from the environment. In situations that lead to a decrease in oxygen tension, a pulmonary vascular response is activated to ensure adequate blood oxygenation [29]. In the lung, alveolar hypoxia causes vasoconstriction of the small pulmonary arteries. The mechanisms responsible for this vasoconstriction depend on pulmonary artery smooth muscle cells (PASMC) and endothelial cells, so the first sensor of oxygen depletion is the pulmonary vasculature [30,31]. In addition, pulmonary vasoconstriction due to hypoxia is mediated by increased intracellular calcium in human PASMC [32]. The availability of oxygen is crucial for cells to carry out different cellular processes, especially the production of energy in the form of adenosine triphosphate (ATP); the mitochondria are the central producer of ATP through metabolic pathways such as the tricarboxylic acid cycle (TCA) and oxidative phosphorylation (OXPHOS) [33,34]. When the demand for oxygen at the cellular level exceeds its supply, the cell enters a critical metabolic state that requires a strategic shift in its metabolism to adapt to low oxygen concentrations and maintain tissue survival (Figure 1) [35].

Under normoxic conditions, the most efficient pathway for obtaining energy is glucose oxidation through glycolysis, TCA, and OXPHOS. Mitochondria are the organelles that consume the most significant amount of oxygen. Oxygen is used in OXPHOS, and this pathway, in turn, is the primary producer of ATP for the maintenance of cellular processes [36]. The normal distribution of oxygen in the tissues is essential to maintain homeostasis and results from an adequate balance between supply and consumption [37]. Typical oxygen pressure values specific to each organ or tissue are called tissue normoxia or physioxia [38]. 

A hypoxic condition refers to a decrease in oxygen concentration below what a cell requires to perform its functions optimally [39]. The reduction in oxygen availability activates several signaling pathways, which trigger transcriptional and metabolic responses to maintain cellular homeostasis [34]. In mammals, physiological hypoxia is related to adequately activating embryogenesis, wound repair, and maintaining the pluripotential of stem cells [40]. In contrast, pathological hypoxia can cause cell damage; increased altitude can produce this state, as well as decreased blood supply to a tissue due to tumors that contain hypoxic zones [40,41].

Cells in different tissues have variable sensitivity to oxygen and possess different ranges of tolerance to hypoxia. The brain is one of the most sensitive and poorly tolerant organs to a decrease in oxygen levels; brain tissue will not survive beyond three minutes of oxygen deprivation, and the level required by this organ is 4.6% oxygen (35 mmHg) [42,43]. The most tolerant tissue to oxygen deprivation is the kidney, which will survive for up to 15–20 min in a hypoxic state [43]. The level of this organ is 9.5% oxygen (72 mmHg) [44]. Therefore, we should be careful using the term “normoxia” and consider the more accurate physioxia instead.

## 3. Hypoxia Changes Mediated by Hypoxia-Inducible Factors

The most critical transcriptional regulators of the response to hypoxia are the HIFs. This family of transcription factors includes three alpha isoforms (HIF-1α, HIF-2α, and HIF-3α) and one beta isoform (HIF-1β) which regulate multiple target genes, including genes involved in metabolic pathways and cell survival, including TCA and OXPHOS, allowing cells to adapt to a hypoxic state [45,46,47].

The functional structure of the HIFs is a heterodimer formed by an alpha and a beta subunit, which have a basic helix-loop-helix (bHLH)-per RNA Sim (PAS) domain important for DNA binding [48,49], an oxygen-dependent degradation domain (ODDD), and two N-terminal (NAD) and C-terminal (CAD) transactivation domains, the latter of which is located in the TAD domain [50]. The structure of the HIF-1α and HIF-2α isoforms is very similar, but some differences are observed; for example, variations in their ODDD domain positioned in the N-TAD region, which contains specific proline residues. In the case of HIF-1α, this is at position Pro402 and Pro564, while HIF-2α contains residues at position Pro405 and Pro531 [51]. HIF-2α is a protein with 48% amino acid parity with HIF-1α; this protein is regulated in the same way by prolyl-hydroxylation; it also dimerizes with HIF-1β and binds to the same hypoxia response element (HER) [52]. HIF-1α is dependent on oxygen, so under normoxic conditions, it is hydroxylated by two types of oxygen-dependent dioxygenase enzymes, proline-dependent prolyl hydroxylases (PHDs), and the inhibitory enzyme of factor HIF directed at asparagine (FIH) [53]. The hydroxylation of HIF-1α is carried out by these enzymes, catalyzing the oxidative decarboxylation of 2-oxoglutarate or α-ketoglutarate, obtaining carbon dioxide (CO_2_) succinate as products. Once the hydroxylation of the HIF-1α prolines occurs, it is recognized by the von Hippel Lindau protein (pVHL), exposing the recognition site for ubiquitin ligase, then HIF-1α is polyubiquitinated and subsequently degraded by the proteasome in the cytosol (Figure 2) [53,54,55,56]. In contrast, when the cell is in hypoxia, the hydroxylase activity of PHDs and FIH is inhibited, blocking the ubiquitination site, allowing the accumulation of HIF-1α in the cytosol and its translocation to the nucleus. Here, it heterodimerizes with the HIF-1β subunit, which will enable it to recognize and bind to a consensus sequence (5′-RCGTG-3′) called HER, which allows it to recruit transcriptional coactivators to activate transcription in genes in response to hypoxia (Figure 2) [56]. These transcriptional genes are involved in a wide range of adaptive responses, mainly focused on the upregulation of transcriptional cascades that are important for protecting and adapting tissues to this condition [56]. In a normoxic state, the α-subunit is constitutively expressed but quickly degraded (Figure 2) [57].

In hypoxic conditions, ATP is obtained through glycolytic metabolism, transforming glucose into pyruvate and then this into lactate through anaerobic fermentation thanks to the activity of the enzyme lactate dehydrogenase (LDH), where two ATP molecules are obtained for each molecule of oxidized glucose [33,34]. Although proteasomal degradation is prevented under these conditions and HIF-1α is stabilized, it causes, among other things, an increase in the transcriptional activity of pyruvate dehydrogenase kinase (PDK), and this triggers inhibition of the enzyme pyruvate dehydrogenase (PDH) and a decrease in the production of acetyl-CoA, favoring increased activity of lactate dehydrogenase (LDH) enzyme and lactate production. All of the above leads to the suppression of TCA and OXPHOS activity to reduce oxygen consumption at the mitochondrial level and maintain ATP production by the anaerobic route [57,58]. HIF-1α transcriptional activity also affects glutamine catabolism by reducing its carboxylation for intermediate formation, and thus its entry into TCA [59]. Thus, in hypoxia, a reduction in glutamine metabolism, acetyl-CoA formation, and lipid synthesis has been observed, mainly associated with the enzymatic activity of isocitrate dehydrogenase 1 (IDH-1), although the mechanisms have not yet been entirely determined [60]. Hypoxia alters this pathway, resulting in a reduction in ATP production and an incremental increase in reactive oxygen species (ROS) production. ROS overproduction is usually limited through the activity of the electron transport chain (ETC) located in the mitochondria, where up to 90% of the cellular oxygen consumption is used by cytochrome c oxidase (COX) when converting H_2_O_2_ into water [61].

The higher level of ROS alters several mechanisms, including electron flow because of reduced ETC activity, reverse electron flow activity in mitochondrial complex I, lack of substrates for oxidative phosphorylation, oxygen imbalance, electron flow imbalance, and inability to dismutate free radicals [36,62,63]. Some of the alterations of OXPHOS in hypoxic conditions are due to changes in the components of the mitochondrial complexes that participate in this pathway; many of the mitochondrial adaptations to hypoxia are untimely regulated by the activity of HIF-1α [36]. The formation of an electrochemical proton gradient is required for the reduction of molecular oxygen; an essential part of the driving forces of ETC is the consumption of oxygen by COX 4. In normoxia, this complex has a COX 4-1 subunit that increases COX activity, whereas in hypoxic conditions, HIF-1α signaling counteracts ROS production by its transcriptional activity, inducing an isoform change from the COX 4-1 subunit to the subunit COX 4-2, which is an isoform that allows electron transfer and oxygen consumption to occur more efficiently [64].

## 4. Hypoxia in Physiological Processes

### 4.1. Lung Development

The three isoforms of HIF participate in a relevant way in the development of the lung at the fetal level. In HIF-1α knockout mice cardiac and vascular malformations and embryonic lethality emerge, while in embryos of HIF-2α knockout mice, vascular defects are observed in the embryo and yolk sac [65]. In addition, the blood vessels fuse incorrectly or do not assemble [66]. As observed in HIF-3α knockout mice, this factor participates in the morphogenesis of late branching, alveolarization, and lung epithelial differentiation [67].

Angiogenesis is another physiological process in which hypoxia is relevant. As a potential initiator, hypoxia regulates the expression of proangiogenic molecules, and though the mechanisms by which this process is carried out are complex, these include the transcriptional activity of HIFs [68]. The HIF-1α and 2α isoforms are known to be homologous and functionally equally involved in angiogenesis; however, there are some differences since HIF-1α regulates and promotes proliferation and migration in early angiogenesis, while HIF-2α participates in remodeling and microvasculature, controlling vascular morphogenesis, integrity, and assembly [69]. The expression of growth factors such as vascular endothelial growth factor (VEGF) is dependent on HIFs, where HIF-2α regulates the level of the VEGFR-2 receptor and participates in endothelial differentiation and angiogenesis [70], and HIF-1α regulates the VEGFR-1 receptor [71]. The induction of angiogenesis can be regulated at the transcriptional level by activating HERs through HIFs; however, in some cases, this induction does not depend on the activation of HERs but on hypoxic conditions or overexpression of HIF-1α through the activation of some transcriptional cofactors [68]. For example, the expression of the proangiogenic protein VE-cadherin, which is regulated by HIF but not by hypoxia, depends on the activation of its promoter, which contains multiple HERs that bind with endothelial nuclear factors. However, the expression of this protein does not depend on hypoxia but on the activation of its promoter through the transcriptional activity of ETS-1, which is dependent on HIF-2α and not on HIF-1α [59]. The nitric oxide synthase gene involved in angiogenesis has an HER promoter. Unlike the previous one, it requires a hypoxic environment for its activation and is also stimulated by HIF-2α [72].

### 4.2. Cell Cycle

In addition, another biological process where hypoxia and HIFs are intricated is the cell cycle; this process is a set of events through which the cell can duplicate its genome, grow, and divide. It consists of four phases: growth (G1), synthesis (S), a second growth period (G2), and the division or mitosis phase (M) [73]. The cycle is regulated by a set of serine/threonine-type protein kinases known as cyclin-dependent kinases (CDKs) and by the oscillation of other cyclin proteins, which form complexes and are activated and inactivated by phosphorylation and dephosphorylation of their substrates to generate the transition from one phase to another during the cell cycle [73]. The process has a high energy demand and can be affected by external stimuli, such as decreased oxygen availability. The oxygen sensing system impacts the cell cycle by stimulating cell proliferation or arrest [74,75].

Oxygen is an essential molecule for eukaryotic cells, as it is the primary substrate for producing biochemical energy in the form of ATP. Cellular processes have a high energy demand for the maintenance of cellular homeostasis. Under hypoxic conditions, the cell enters a critical metabolic state that requires a strategic shift in its metabolism to adapt to low oxygen concentrations and maintain tissue survival. Regarding cell cycle regulation complexes, Myc is an oncogene that encodes the Myc oncoprotein with the transcriptional and non-transcriptional activity that can induce or repress the expression of genes involved in DNA replication, RNA processing, cell differentiation, division, and apoptosis (Figure 3A) [76].

Hypoxia has a complex relationship with the cell cycle since it can stimulate or inhibit cell growth, depending on the cell type, the microenvironment, or the cell signaling pathways activated in the cells. For example, hypoxia is essential in response to wounds and injuries and is related to the stimulation of cell growth, because the damaged tissue must be replaced. However, in cells not related to this mechanism, the classic cellular response to hypoxia is growth inhibition [77]. In tumors, critical genes involved in the progression and regulation of the cycle are modified, and here HIFs participate together with other proteins as cofactors in the mechanisms related to the inhibition or stimulation of cell division (Table 1). For example, the expression of HIF-1α tends to functionally counteract Myc by inhibiting MYC/MAX complex formation, even without hypoxia (Figure 3B) [78]. Additionally, cyclin-dependent kinase inhibitors p27 and p21 are induced by hypoxia, which inhibits CDK2 activity and prevents entry into S phase through hypo-phosphorylation of retinoblastoma protein [77,79], generating cell cycle arrest in the cell. HIF-1α, through interference with the ATR activating protein, leads to a negative effect on DNA replication and promotes cell arrest [80,81]. HIF-1α has transcriptional activity in different proteins and genes. It also has significant activity in microRNAs, such as miR210, which is upregulated in a HIF-1α-dependent manner, and has an inhibitory effect on cell cycle progression in Human Pulmonary Artery Endothelial Cells (Table 1) [81].

In contrast to the activity of HIF-1α against cell division, it has been observed that despite the structural similarities between HIF-1α and HIF-2α, the activity of the latter isoform is opposite to that of HIF-1α in the cell cycle; i.e., HIF-2α has positive transcriptional activity on the cell cycle, promoting cell proliferation this is possible because it helps to stabilize the Myc/MAX complex (Figure 3B). When this complex is stable, it binds to DNA promoter sites and stimulates cyclin D1 activity and thus cell proliferation, this has been observed in renal carcinoma cells, NTH3T3 cells, HEK293 cells, and embryonic epithelial cells. In addition, HIF-2α has its specific target gene, Oct-4, a transcription factor necessary to regulate the differentiation and function of stem cells and maintain their pluripotent character [73]. Additionally, an enrichment analysis of Myc in hypoxia showed a response in several pathways [82]. In this sense, dimers of Myc and HIF-2α induced the proliferation of hepatocellular carcinoma cells in a mild chronic hypoxia model; this supports the hypothesis that the PI3K/mTORC2/HIF-2α/c-Myc axis may play a vital role. Therefore, the PI3K inhibitor apitolisib may serve as a possible treatment option for patients suffering from this type of tumor, especially in cases with rapidly growing tumors under mild chronic hypoxic conditions [83].

**Table 1 cells-11-02938-t001:** Effect of HIF-1α and HIF-2α on the cell cycle.

Gene	TF	Cell Cycle	Finding	References
c-Myc	HIF-1α	Arrest	Prevents the formation of complexes of Myc with its promoters and therefore the activation of its target genes	[78]
	HIF-2α	Proliferation	Promotes Myc binding to its promoters and activation of its target genes	[73]
p27	HIF-1α	Arrest	HIF-dependent induction in lymphocytes by displacement of Myc to its promoters	[73,78,79]
p21	HIF-1α	Arrest	HIF-dependent induction in fibroblasts by displacement of Myc to its promoters.	[78,79]
Cyclin D2	HIF-1α	Arrest	Prevents the formation of Myc-DNA binding site complexes and alters the expression of Cyclin D2	[78]
ATR	HIF-1α	Arrest	Interferes with ATR activating protein and promotes ATR activation	[80]
oct-04	HIF-2α	Differentiation	Regulation of cell differentiation in stem cells	[73]
miRNA 210	HIF-1α	Arrest	HIF-1α-dependent regulation	[81,84]
AURKA	HIF-1α	Proliferation	Cell proliferation hepatocellular carcinoma	[85]
		Decreases AURKA activity	Negative regulator of AURKA in breast cancer tumors	[86]

### 4.3. Immune Response

HIFs have multiple functions in immune cells and vary according to the context and environment in which they are activated (Table 2) [87]. The tissue environment is hypoxic during a bacterial infection, and some infections can induce HIF-1α activation even in normoxia. In inflammation, HIF-1α and HIF-2α have differential responses; for example, in the liver, HIF-2α can directly activate inflammatory mediators. Il-6 has been shown to be a direct target gene of HIF-2α in macrophages, and HIF-1α can regulate the production of molecules related to antimicrobial activity, such as proteases, antimicrobial peptides, and nitric oxide (NO) [88].

In macrophages, they play a determining role in antimicrobial capacity [84,89,90]. HIF-1α deletion in myeloid lineages promotes an inappropriate inflammatory response associated primarily with disorders of the glycolytic pathway, which decreases energy production and impairs macrophage aggregation, invasion, and motility [91]. Furthermore, it has been shown that hypoxia, as well as the transcriptional activity of HIF-1α and the induction of the glycolytic pathway, are necessary for the development, differentiation, and proliferation of Th17, Treg, and CD8+ T cells [92,93]. HIF-1α deficiency in B cells results in abnormal B cells; the development of these cells is dependent on glycolysis because they require further induction of the HIF-1α-dependent glycolytic pathway [94,95].

In light of the above, the response to hypoxia-mediated by HIF plays a critical role in the regulatory activity of the innate and adaptive immune response, in addition to being related to multiple inflammatory diseases [87,96].

**Table 2 cells-11-02938-t002:** Role of HIF-1α and HIF-2α in the immune response.

Cell Type	HIF Activity in the Immune Response	Reference
Bacterial infections	Control of the intracellular antibacterial response by macrophages by HIF-1α	[91]
	Control of bacterial phagocytosis	[89]
	HIF-1α-dependent antimicrobial activity in myeloid cells through nitric oxide expression	[88]
Macrophages	Regulation of macrophage motility, invasion, and aggregation by HIF-1α	[91]
	Polarization of M1 macrophages by HIF-1α activity secondary to TH1 induction and of M2 macrophages by HIF-2α induced by Th2 cells	[97]
	Modulation of macrophage migration by HIF-2α regulatory activity of cytokine receptor expression	[98]
Neutrophils	mTOR regulates NET formation by transcriptional control of HIF-1α expression in hypoxia	[99]
	Reversible inhibition of neutrophil apoptosis by hypoxia, could be related to HIF-1α activity	[100]
	HIF-2α regulates neutrophil apoptosis in vivo, reducing inflammation and tissue injury	[101]
Dendritic cells	HIF-1α and hypoxia play a role in the activation of dendritic cells in an inflammatory state	[102]
	Increased migratory capacity of dendritic cells and HIF-1α-dependent induction of IL-22 in hypoxia	[103]
	Pharmacological certainty of HIF-1α by PDH inhibitor increases MHC, co-stimulation of molecule expression and reduction of T cells	[104]
	HIF-1α activity on migration of dendritic cells matured in hypoxia	[105]
Chemokines cytokines	Regulation of expression of M-CSFR cytokine receptors and CXCR4 chemokines	[98]
T cells	HIF-1α-dependent glycolytic metabolic switch is a checkpoint for Th17 and Treg cell proliferation	[106]
	HIF-1α is involved in downregulation of Th1 cells	[92]
	HIF-1α is required for the regulation of glycolytic pathways, chemokine expression, and adhesion receptors that regulate CD8+ T cell trafficking	[93]
B Cells	HIF-1α activity in the glycolytic pathway affects B cell development and differentiation	[95]
	HIF-1α has transcriptional activity in IL-10 expression in CD1dhiCD5+ B cells and in the control of its protective activity in autoimmune diseases	[107]

### 4.4. Regeneration and Repair

Under normal conditions, the repair and regeneration processes are highly coordinated by a hierarchy of signaling pathways, where HIF has a significant role [108]. In a regeneration model (*Hemidactylus platyurus*), the expression of HIF-1α reached its peak on the third day, thereafter decreasing. At this point, the expression of HIF-2α gradually increased and was maintained for thirteen days. Furthermore, in the MRL mouse regeneration model, systemic levels of HIF-1α were enhanced after an injury, peaking between days 10 and 14, and subsequently decreasing the next month throughout regeneration. This biphasic response correlates well with inflammation and remodeling, consistent with a dedifferentiation pattern followed by cellular re-differentiation [109].

Hence, this reflects that the expression kinetics of HIF-1α and HIF-2α in regeneration are sequential, where in the initial stage HIF-1α participates, whereas HIF-2α participates in the final stage. This pattern is similar to the one mentioned in the section related to differential expression of HIFs, where HIF-1α participates in acute hypoxia and HIF-2α in chronic hypoxia [110]. Regarding the tissue and intracellular localization of HIF throughout the regeneration period, it was observed that on the first day, HIF-1α was present in the nuclei of the basal lamina cells in the dermis layer. On the third, it spread to nuclei of fibroblast-like cells and, on day five, to nuclei of ganglion cells [110]. In contrast, HIF-2α was found in the nuclei of peripheral nerves and similar cells on the third day, and on day ten, it was extended to the nuclei of endothelial cells [110]. This distribution allowed us to clearly define that the two isoforms participate in a complementary manner with different timings in the tissue regeneration process.

## 5. Hypoxia-Inducible Factor-2α has a Particular Response in Idiopathic Pulmonary Fibrosis Pathogenesis

The pathological pattern of Usual Interstitial Pneumonia (UIP), characteristic of IPF, is heterogeneous, with areas of pulmonary fibrosis manifesting with foci of fibroblastic proliferation caused by epithelial damage and activation. These foci are located in the pulmonary interstitium and are characterized by the proliferation of fibroblasts and myofibroblasts, along with decreased apoptosis and hyperreactivity to fibrogenic cytokines [111]. Fibroblasts have a greater response capacity against profibrogenic cytokines, such as via transforming growth factor beta-1 (TGF-β1); this factor is involved in cellular functions such as cell proliferation, differentiation, and apoptosis [112], and the accumulation and activity of HIF-1α even under normoxic conditions, which in turn induces the expression of VEGF. This effect is enhanced by TGF-β1, which additionally inhibits the expression of the PHD2 gene through the Smad signaling pathway [113].

Fibroblasts constitute a diverse population of cells whose primary function is to establish, maintain, and modify the connective tissue stroma. In addition to interacting with various tissues, their primary function is to secrete proteins that constitute the extracellular matrix and play an essential role in wound repair, tissue development, and fibrosis [111]. In IPF, the population of pulmonary fibroblasts manifests a pathological phenotype, which may cause the perpetuation of this disease [114,115,116,117,118,119,120]. It has been suggested that activated fibroblasts acquire an aggressive phenotype and are the primary inductor of the accumulation of collagen in IPF [121]. Among the factors that induce this activation, we here highlight the role of TGF-β1 and hypoxia. Both induce the differentiation of fibroblasts to myofibroblasts, protection against apoptosis, and expression of α-actin in its cytoskeleton. In addition, this cytokine triggers the phenomenon known as Epithelial-Mesenchymal Transition (EMT) [122]. Notably, hypoxia promotes the proliferation in lung fibroblasts of patients with IPF, as well as in healthy subjects [123,124]. Significantly, hypoxia or another factor, such as TGF-β1, can promote the stabilization and activation of HIF-1α, favoring the production of LDH-5 not only in fibroblasts but also in the lung epithelium. These stimulate the differentiation of fibroblasts to myofibroblasts, forming a vicious cycle where hypoxia causes fibrosis and this, in turn, further promotes hypoxia [125].

HIFs are the main transcription factors that regulate the hypoxia response, so it is relevant to emphasize the structural differences of the HIF-1α and HIF-2α isoforms in specific proline residues, and the impact of those differences on the hypoxia response and IPF. An increasingly accepted hypothesis postulates that HIF-1α participates in regulating genes involved in the response to acute hypoxia, while HIF-2α has functions in chronic hypoxia [126,127]. In this context, hypoxia is acute when oxygen deprivation is for a short period, ranging from 2 to no more than 24 h, with an oxygen concentration of less than 1% [127,128]. The response to this type of hypoxia is mediated mainly by HIF-1α, which represents an initial response to hypoxia. In contrast, chronic hypoxia is hypoxia lasting more than 24 h, with an oxygen concentration of less than 5% [128]. In some studies, it has been observed that there is a change in the stabilization of HIF isoforms in neuroblasts and astrocytes during chronic exposure to a hypoxic environment, so this type of behavior supports the notion that the response to chronic hypoxic conditions is mainly mediated by HIF-2α [128,129]. HIF-1α has been reported to bind near the promoters, while HIF-2α binds to distal enhancers, and their distribution is not affected by the degree or duration of hypoxia or cell type. Additionally, the two isoforms do not compete for binding sites [130]. HIF-2α binds to distal enhancers, which could have functional repercussions that depend on other co-regulatory genes; for example, inhibition of HIF-2α function is produced by the recruitment of transcription cofactors or co-repressors in promoters of endogenous target genes [131].

We hypothesize that a delicate balance determines whether a tissue can activate the regeneration process or induce fibrosis, depending on the dysregulation and overexpression of HIFs. This work highlights the role of HIF-2α in the fibrogenic process. Recently, the dysregulated expression of HIF-2α has been proposed as a determining factor in the development of fibrosis; therefore, its overexpression in pulmonary fibroblasts from patients with IPF is probably the cause of the pseudohypoxia phenotype (or aerobic glycolysis) characteristic of these cells [22,132]. On the one hand, there is evidence that supports a correlation between hypoxia and the proliferation of IPF fibroblasts in a HIF-2α-dependent manner [21]. Genes that control cell proliferation and growth are regulated through HIF transcriptional factors [133]. Studies have shown that silencing of HIF-2α correlates with decreased miR-210 levels, leading to lower proliferation in fibroblasts derived from the lungs of IPF patients cultured in a hypoxic environment. In contrast, when HIF-1α is blocked, proliferation is not affected [21]. On the other hand, HIF-2α exclusively regulates some genes involved in VEGF activation in endothelial cells [52,134], correlating with the respiratory distress syndrome (RDS) observed in newborn mice caused by the loss of HIF-2α, which results in low production of surfactant by type II pneumocytes; the administration of VEGF prevents this syndrome [135]. Hypoxia increases VEGF-A expression in monocytes, fibroblasts, keratinocytes, myocytes, and endothelial cells [136,137,138,139]. This incremental increase in the production of VEGF-A in the different cell types that participate in regeneration and wound healing processes could be via activation of HIF-2α during hypoxia. In a previous study, we observed that the expression of α-smooth muscle actin (fibroblast to myofibroblast differentiation marker) resembles the expression pattern of HIF-2α, since both genes are overexpressed after 24 h of hypoxia exposure and are sustained for up to 96 h [22].

Recently, the HIF-1α signaling pathway has been identified as an essential factor in wound healing [140], and it is suggested that decreased regeneration could be related to changes in fibroblast activity [141]. HIF transcriptional activity has been reported to enable fibroblast to myofibroblast differentiation and the production of profibrotic mediators, where HIF signaling acts as an amplifier of IPF [56,113,142]. However, we consider that according to the evidence generated in animal models, regeneration depends on the regulation of oxygen by HIFs and, in turn, these promote a particular adaptation that affects the tissue repair process [109,110,143,144,145,146,147]. We recently proposed that hypoxia signaling pathways only make sense in the context of lung regeneration. Given that pulmonary fibrosis is assumed to be a product of damage and that the lung is trying to repair or regenerate itself, the hypoxia response would be necessary for the regeneration process; however, if it persists, it can lead to the activation of related feedback loops with disease progression [148]. More research is needed to elucidate the mechanisms involved in tissue regeneration.

After tissue damage, a regeneration or scarring process can occur, but the determinants are still unclear; one of the reasons for this is probably due to evolutionary processes. Through animal regeneration models, HIFs have been determined to participate in these mechanisms. Evidence exists to reaffirm that fibrosis is an aberrant regeneration process that does not end. The kinetics of the regeneration process involves HIF-1α in the initial stage, HIF-2α in the intermediate stage, and HIF-3α in the final stage of regeneration. This does not happen in the same way in fibrosis, since multiple studies have indicated that in lung fibroblasts from patients with IPF and in other tissues with fibrosis, high levels of HIF-1α and HIF-2α are maintained (Figure 4). At the same time, HIF-3α is significantly decreased during hypoxia compared to healthy tissue or healthy lung fibroblasts in normoxia; therefore, we assume that this alteration does not allow a proper regeneration to be completed, and fibrosis persists.

## 6. Conclusions

Hypoxia and dysregulation in HIF kinetics are involved in a specific adaptation to the pathophysiology of IPF. It is already known that hypoxia can be part of the activation of fibroblasts towards a profibrotic loop. However, the role of epithelial cells and hypoxia remains undiscovered. Here, we propose that HIF-2α and hypoxia are associated with chronic and persistent regeneration, leading to the perpetuation of fibrosis.

## Figures and Tables

**Figure 1 cells-11-02938-f001:**
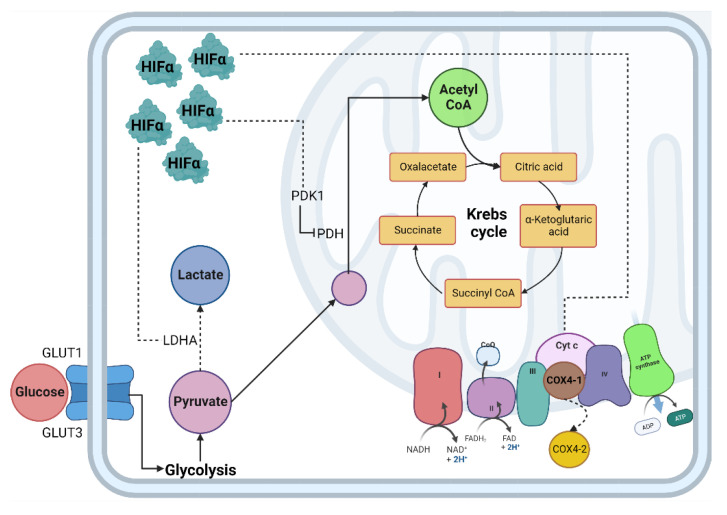
Cellular adaptation to hypoxia. The primary method of obtaining energy is the oxidation of glucose; under normoxic conditions, this begins in the cytosol through glycolysis, followed by the decarboxylation of pyruvate, then the TCA cycle where NADH and FADH2 are obtained, which, by giving up their electrons to the chain transport electrons, create an electrochemical proton gradient in the mitochondrial intermembrane and allow ATP synthase to release ATP molecules. In hypoxia, the enzymes involved in the glycolytic pathway, such as PDH, are inhibited by PDK1, decreasing the production of acetyl-CoA and increasing the production of lactate due to the activity of the enzyme LDH. Abbreviations: TCA: Tricarboxylic acid; NADH: reduced nicotinamide adenine dinucleotide; FADH2: reduced flavin adenine dinucleotide; ATP: adenosine-5’-triphosphate; PDH: pyruvate dehydrogenase; LDH: lactate dehydrogenase; HIF-α: Hypoxia Inducible Factor α subunit; HIF-β: Hypoxia Inducible Factor β subunit; Dotted line: HIF-α mitochondrial target gene.

**Figure 2 cells-11-02938-f002:**
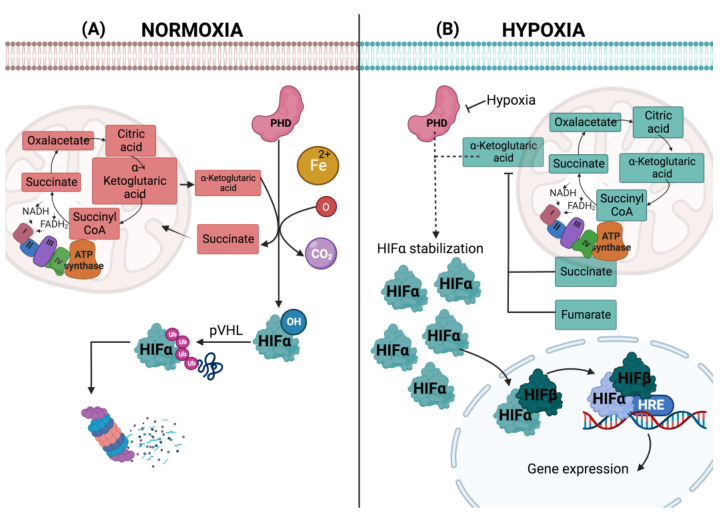
Hypoxia response mediated by HIFs. In normoxia, the hydroxylation of HIF- occurs in the cytosol, where PDH catalyzes the oxidative decarboxylation of α-ketoglutarate, obtaining carbon dioxide (CO_2_) and succinate once HIF- is hydroxylated and polyubiquitinated, a reaction catalyzed by pVHL. Finally, HIF- is degraded by the proteasome in the cytosol (**A**). In hypoxia, instead of being degraded by the proteasome, HIF- stabilizes in the cytosol and translocates to the nucleus. It heterodimerizes with HIF-1β, and both subunits bind to HER, allowing it to recruit transcriptional coactivators to activate the transcription of hypoxia response genes (**B**).

**Figure 3 cells-11-02938-f003:**
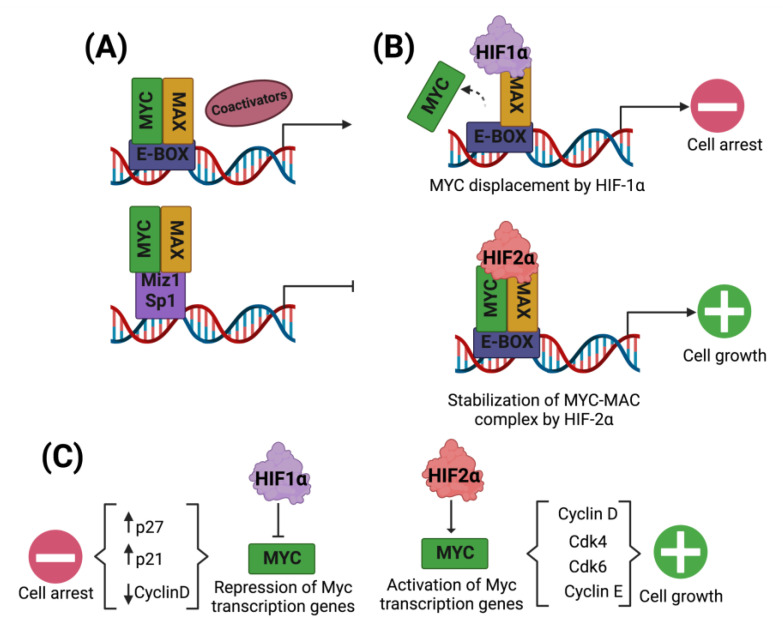
HIF regulation of the Myc/MAX complex in the cell cycle. Myc transcriptional gene activations are regulated by Myc dimerization with the Max protein and binding to its promoters and DNA coactivators. When the Myc/MAX complex does not bind promoters and instead binds Miz1 and Sp1, the transcription of target genes for Myc is inhibited (**A**). In addition, HIF-1α displaces Myc, inhibiting the Myc/MAX complex and suppressing transcription of Myc-regulated genes, leading to inhibition of cell proliferation. On the contrary, HIF-2α promotes the Myc/MAX complex stabilization, favoring cell proliferation (**B**). HIF-1α represses the transcription of Myc target genes, increasing p27 and p21, leading to cell proliferation arrest. On the other hand, HIF-2α promotes cell proliferation by activating the transcription of target genes for Myc, which stimulates the activity of cyclins (**C**).

**Figure 4 cells-11-02938-f004:**
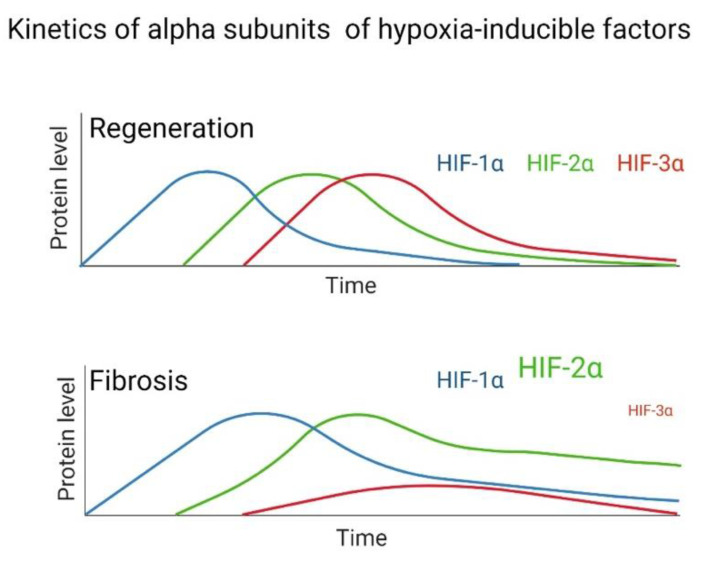
Hypothesized difference in the kinetic pattern of HIFs between a fibrotic and regenerative model during hypoxia.

## Data Availability

Not applicable.

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
