# Peer review of "Functional Repercussions of Hypoxia-Inducible Factor-2α in Idiopathic Pulmonary Fibrosis"

_cells, 2022, doi:10.3390/cells11192938_

Round 1

Reviewer 1 Report

 The manuscript ‘Functional Repercussions of Hypoxia-Inducible Factor-2α in 2 Idiopathic Pulmonary Fibrosis’ by Torres-Soria and Romero et al, is a descriptive review on the importance of of HIF2a in the IPF. The manuscript looks well written. There is good description of the role of metabolism, hypoxia, and HIF transcription factors in general. However, the review is lacking detailed description of the HIF in IPF, as the title says. There seems a lack of focus on the topic intended for the review, but more on introductory topics.  

Major comments 

  1. Little information is presented describing HIF and hypoxia in IPF disease. 
  2. Several pieces of literature not discussed e.g. Barratt et al, 2018, Porter et al, 2021, Burman et al, 2018. 

Specific comments 

  1. Figure 1, check English 
  2. Check headings 
  3. Line 152, check references 
  4. Line 648, check grammar 

Author Response

Reviewer #1

The manuscript ‘Functional Repercussions of Hypoxia-Inducible Factor-2α in 2 Idiopathic Pulmonary Fibrosis’ by Torres-Soria and Romero et al, is a descriptive review on the importance of HIF2a in the IPF. The manuscript looks well written. There is good description of the role of metabolism, hypoxia, and HIF transcription factors in general. However, the review is lacking detailed description of the HIF in IPF, as the title says. There seems a lack of focus on the topic intended for the review, but more on introductory topics.  

Major comments 

  1. Little information is presented describing HIF and hypoxia in IPF disease. 

R1. We agree with the reviewer, we reorganize IPF information throughout the paper, and we add additional information on IPF in this revised version of the manuscript.

  1. Several pieces of literature not discussed e.g. Barratt et al, 2018, Porter et al, 2021, Burman et al, 2018. 

 R2. Thanks for this comment. References were added in the introduction section of the revised manuscript.

Reviewer 2 Report

In this manuscript, Ana Karen Torres-Soria et al. have collected information and described the involvement of hypoxia-inducible factors (HIFs) in the processes associated with the development of idiopathic pulmonary fibrosis (IPF) and finally focus on the dysfunctional involvement of HIF-2α in the mechanisms of lung regeneration and repair, proposing HIF-2α as the main or one of the main responsible for the aberrant tissue regeneration processes characteristic of IPF.

The manuscript has its merits and is very relevant to scientific interest. However, I consider that it has some deficiencies that can be corrected if the following points are considered.

  1. It is recommended to read the text thoroughly (grammar and syntax) and check when using abbreviations (in the text, figures, and tables) for the first time.
  2. In Figure 1, page 4, there is a word in Spanish (Ciclo de).
  3. The description of HIF in the manuscript is not very clear since it can be thought that is described is in general processes or fibrotic processes, or specifically in IPF; therefore, it is suggested to explain in which method it is participating in which tissues and cells, as well as the conditions (physiological or pathological). 
  4. In section 10, 11, 12, 13, and 14, it is not clear if it refers directly to IPF or to fibrosis or if it is in general; it is suggested to look for more evidence or to be more precise. 
  5. Section 15 seems to be the only one that talks about HIF applied to IPF; I think that this point can be expanded since I believe that it is too little information for the title of the manuscript to refer to IPF.
  6. The proposed title is interesting; however, almost all of the text is focused on describing the role of hypoxia in physiological states and its repercussions in the development of fibrosis, and the role of HIF 2 alpha in fibrosis is discussed in sections 14 and 15, a title that would be appropriate to the content of the manuscript can be suggested.
  7. The conclusions are very ambiguous; the findings are too general, without a clear result and an adequate description of prospects.

Author Response

Reviewer #2

In this manuscript, Ana Karen Torres-Soria et al. have collected information and described the involvement of hypoxia-inducible factors (HIFs) in the processes associated with the development of idiopathic pulmonary fibrosis (IPF) and finally focus on the dysfunctional involvement of HIF-2α in the mechanisms of lung regeneration and repair, proposing HIF-2α as the main or one of the main responsible for the aberrant tissue regeneration processes characteristic of IPF.

The manuscript has its merits and is very relevant to scientific interest. However, I consider that it has some deficiencies that can be corrected if the following points are considered.

  1. It is recommended to read the text thoroughly (grammar and syntax) and check when using abbreviations (in the text, figures, and tables) for the first time.

R1. We apologize by several errors found in the previous version.  

  1. In Figure 1, page 4, there is a word in Spanish (Ciclo de).

R2. We corrected this error in the revised version.

  1. The description of HIF in the manuscript is not very clear since it can be thought that is described is in general processes or fibrotic processes, or specifically in IPF; therefore, it is suggested to explain in which method it is participating in which tissues and cells, as well as the conditions (physiological or pathological). 

R3. Following this recommendation, we highlight the IPF in the introduction section of the manuscript.

  1. In section 10, 11, 12, 13, and 14, it is not clear if it refers directly to IPF or to fibrosis or if it is in general; it is suggested to look for more evidence or to be more precise. 

R4. This is an important comment. We highlight the IPF information throughout the paper in this revised version of the manuscript.

  1. Section 15 seems to be the only one that talks about HIF applied to IPF; I think that this point can be expanded since I believe that it is too little information for the title of the manuscript to refer to IPF.

R5. We agree with the reviewer, IPF information and references were added now in this revised version.

  1. The proposed title is interesting; however, almost all of the text is focused on describing the role of hypoxia in physiological states and its repercussions in the development of fibrosis, and the role of HIF 2 alpha in fibrosis is discussed in sections 14 and 15, a title that would be appropriate to the content of the manuscript can be suggested.

R6. Thanks for this suggestion, we believe that since this version was restructured the title can be kept.

  1. The conclusions are very ambiguous; the findings are too general, without a clear result and an adequate description of prospects.

R7. We agree with the reviewer, we rewrite the conclusions in the revised version of the manuscript.

Round 2

Reviewer 1 Report

The manuscript still has errors:

1. Line 397, what is VEG? Is the reference appropriate?

2. Line 412, does hypoxia promote proliferation in humans or human cells? This statement is confusing. 

3. The manuscript in multiple places is repetitive and confusing.

4. The introductory part on hypoxia is still very long. The authors should try to focus more on IPF. 

5. The manuscript will significantly improve if edited by a native English speaker.

Author Response

Dr. Ilja Vietor

Guest Editor of the journal Cells

Dear Ilja

Thank you for allowing us to resubmit our manuscript. We appreciate all comments from reviewers because they helped improve the work. In addition to responding to most of the reviewers' comments, the manuscript was submitted for language review and editing. With these changes, we hope our manuscript can be accepted in your prestigious journal.

Reviewer #1

The manuscript still has errors:

1.- Line 397, what is VEG? Is the reference appropriate?

We are sorry and thank you very much for noticing the error, it was corrected, and the entire bibliography was adjusted in detail to avoid any other error

2.- Line 412, does hypoxia promote proliferation in humans or human cells? This statement is confusing. 

Thank you very much, the wording was changed, and it is as follows:

In particular, hypoxia promotes the proliferation of IPF lung fibroblasts and healthy lung fibroblasts

3.- The manuscript in multiple places is repetitive and confusing.

Thank you very much for this observation. We removed some paragraphs that were repeated.

4.- The introductory part on hypoxia is still very long. The authors should try to focus more on IPF. 

You are right. Thank you very much for the observation; we have eliminated the introductory part of biochemistry and some other paragraph; we have added in the first part of the introduction some fundamental questions about hypoxia in IPF.

Added the following paragraphs:

Idiopathic pulmonary fibrosis (IPF) has a poor prognosis, with a median survival of 24–30 months (1,2), and is characterized by reduced functional capacity, dyspnea, and hypoxia induced by exercise or at rest (3–5). Destruction of lung architecture impairs gas exchange and progresses to hypoxic respiratory failure, a hallmark of advanced disease (2). The blood oxygen saturation level is considered an important parameter because its decrease during endurance tests predicts survival in patients with IPF (6). Most patients have a poor quality of life due to low physical activity and limited exercise tolerance (2,7), decreased lung compliance leading to mechanical ventilation, and increased respiratory muscle energy expenditure, driving dyspnea (8,9).

Exertional dyspnea and worsening hypoxia are clinical features of IPF, and no drug is available to treat these two symptoms (10); even oxygen inhalation does not improve tolerance to physical exertion in most patients and, therefore, does not relieve shortness of breath (11,12). Obstructive sleep apnea (OSA) is known to be a risk factor for IPF; in-termittent hypoxia (HI) and reoxygenation of OSA contribute to a poor prognosis (13). Chronic exposure to HI increases mortality, lung inflammation, and pulmonary fibrosis in BLM-treated mice, suggesting a worse prognosis in patients who have IPF and severe OSA (13,14). Most in vitro studies with cells and in vivo with animal models have shown that hypoxia is a determining factor in the progression and development of the disease. However, at the clinical level, there are still several questions.

Removed the following paragraphs and sentences in section 2 Lung oxygenation and hypoxic conditions

Glycolysis takes place in the cytosol, and through this pathway, two ATP molecules, two nicotinamide adenine dinucleotide (NADH) molecules, and two pyruvate molecules are produced. The NADH glycolysis products donate their electrons through a transport system used in OXPHOS. Pyruvate is metabolized by oxidative decarboxylation of pyruvate to acetyl coenzyme A (acetyl-CoA), which is used as a substrate in TCA (36). Once glycolysis and TCA have been carried out, the electron transporting reducing agents (NADH and FADH2) obtained by these pathways enter an oxidation process in which an electrochemical proton gradient is formed in the mitochondrial intermembrane. Then these protons pass through the inner mitochondrial membrane into the mito-chondrial matrix, they allow ATP synthase to release ATP molecules (35,37,38). The formation of the electrochemical gradient also generates free radicals with the ability to oxidize proteins, lipids, nucleic acids, and the formation of hydrogen peroxide (H2O2). To avoid the damage that H2O2 can cause, the enzyme superoxide dismutase (Mn-SOD) catalyzes the conversion of peroxide to water and oxygen (24–26).

In section 3 Hypoxia changes mediated by Hypoxia-Inducible Factors

To obtain ATP, the oxidation of glucose is required through the before aforementioned pathways;

Multiple pathways converge at the TCA, the metabolites that are part of this cycle are necessary for both anabolic and catabolic processes, so these intermediates are used in anabolic processes and, therefore, must be replenished to maintain ATP production. An example of this is that acetyl-CoA can be obtained from fatty acids and succinyl-CoA from glutamine (62,63). However,

Moreover, the fluctuation of oxygen viability also has an important impact on OXPHOS, as well as on the cellular redox balance (66).

During hypoxia,

As mentioned above

As mentioned before hypoxia is mediated by HIFs and consequently HIFs are involved in some biological processes, for instance, in embryogenesis.

To obtain ATP, the oxidation of glucose is required through the before aforementioned pathways; i

Multiple pathways converge at the TCA, the metabolites that are part of this cycle are necessary for both anabolic and catabolic processes, so these intermediates are used in anabolic processes and, therefore, must be replenished to maintain ATP production. An example of this is that acetyl-CoA can be obtained from fatty acids and succinyl-CoA from glutamine (62,63). However,

As mentioned above,

In section 4 Hypoxia in Physiological Processes

As we mentioned earlier,

5.- The manuscript will significantly improve if edited by a native English speaker.

Answer. - The manuscript has undergone English language editing by MDPI. The text has been checked for correct use of grammar and common technical terms, and edited to a level suitable for reporting research in a scholarly journal. As the reviewer suggested

Reviewer 2 Report

All comments have been addressed 

Author Response

Dear reviewer.
Thank you very much for your contribution to our manuscript.

Regards